# Vertebral Rotation in Functional Scoliosis Caused by Limb-Length Inequality: Correlation between Rotation, Limb Length Inequality, and Obliquity of the Sacral Shelf

**DOI:** 10.3390/jcm12175571

**Published:** 2023-08-26

**Authors:** Martina Marsiolo, Silvia Careri, Diletta Bandinelli, Renato Maria Toniolo, Angelo Gabriele Aulisa

**Affiliations:** 1U.O.C. of Orthopaedics and Traumatology, Bambino Gesù Children’s Hospital, Istituto di Ricerca e Cura a Carattere Sceintifico (IRCCS), 00165 Rome, Italy; silvia.careri@opbg.net (S.C.); diletta.bandinelli@opbg.net (D.B.); renatom.toniolo@opbg.net (R.M.T.); agabriele.aulisa@opbg.net (A.G.A.); 2Department of Human Sciences, Society and Health, University of Cassino and Southern Lazio, 03043 Cassino, Italy

**Keywords:** scoliosis, vertebral rotation, limb inequality, limb discrepancy, sacral shelf obliquity, functional scoliosis, sacral shelf inclination

## Abstract

Background: Scoliosis is a structured rotatory deformity of the spine defined as >10° Cobb. Functional scoliosis (FS) is a curve < 10° Cobb, which is non-rotational and correctable. FS is often secondary to leg length inequality (LLI). To observe vertebral rotation (VR) in functional scoliosis due to LLI, one must demonstrate a correlation between LLI, sacral shelf inclination (SSI), and VR and discover a predictive value of LLI capable of inducing rotation. Methods: We studied 89 patients with dorso-lumbar or lumbar curves < 15° Cobb and radiographs of the spine and pelvis. We measured LLI, SSI, and VR. The patients were divided into VR and without rotation (WVR) groups. Statistical analysis was performed. Results: The mean LLI value was 6.5 ± 4.59 mm, and the mean SSI was 2.8 ± 2.53 mm. The mean value of LLI was 5.2 ± 4.87 mm in the WVR group and 7.4 ± 4.18 mm in the VR group. The mean SSI value for WVR was 1.4 ± 2.00 and that for VR was 3.9 ± 2.39. For each mm of LLI, it was possible to predict 0.12° of rotation. LLI ±5 mm increased the probability of rotation (R2.08 *p* < 0.0016), while this was ±2 mm for SSI (R2 0.22 *p* < 0.01). Each mm of LLI corresponded to 0.3 mm of SSI (R2 0.29, *p* < 0.01). Conclusions: FS secondary to LLI can cause VR, and 5 mm of LLI can cause SSI and rotation.

## 1. Introduction

The term scoliosis, first used by Galen, derives from the Greek word “crooked”. In 1741, André used the crooked spine as his symbol for orthopedics [1]. Scoliosis is a structured deformity of the spine that is expressed in three dimensions of space: a curve in the frontal plane (which is the most evident manifestation) is associated with vertebral rotation in the transverse plane (which is the characteristic element) and a deformity in the sagittal plane. The term “structured” means that one cannot spontaneously correct the curve. Scoliosis can be classified according to its etiology, the location of the curve, and the extent of angular deviation. The most frequent type of pathology is idiopathic scoliosis, with a prevalence of 0.47–5.2%. The prevalence and curve severity are higher for girls than for boys, and the female-to-male ratio increases with increasing age among children. The ratio is 1.5:1 for the mild forms, while it increases by 10:1 for the more severe forms (>30° Cobb). A diagnosis of idiopathic scoliosis is made if a non-idiopathic form has been excluded [2]. Clinically, a patient with scoliosis will present with an asymmetry of the shoulder line, an asymmetry of the size triangles between the line of the arms and that of the hips, and a hump in the anterior bending test. Clinical suspicion should be confirmed based on the presence of a curve of the spine on radiographic examination. The Report of the Terminology Committee of the Scoliosis Research Society, which is the international body responsible for regulating and standardizing the terminology and classifications of vertebral deformities, states that “scoliosis is a lateral curvature of the spine”. According to this definition, deviations of the spine in the frontal plane are generically defined as scoliosis. In the literature, it is universally accepted that for a curve to be defined as scoliosis it must be greater than 10° Cobb in the frontal plane, but there is no agreement on the pathogenesis of idiopathic scoliosis. Various theories have been posited to explain the pathogenesis of idiopathic scoliosis; these can be classified into the following groups to provide a better understanding of the multifactorial pathogenesis of AIS: genetics, mesenchymal stem cells, tissues, spine biomechanics, neurology, hormones, biochemistry, environment, and lifestyle [3,4,5,6]. Despite much research, the mechanism underlying the onset of idiopathic scoliosis remains unknown. Furthermore, the official terminology distinguishes another type of scoliosis in addition to the structured one mentioned above, namely, unstructured scoliosis, also called scoliotic attitude or “functional scoliosis”. This is a mild, non-structural, and steady lumbar curve often secondary to limb length inequality without vertebral rotation. The major skeletal reactions or adaptations to leg length discrepancy are pelvic obliquity and scoliosis [7,8,9]. Leg length inequality (LLI) or discrepancy is a difference between the length of the legs and is a common orthopedic condition with a prevalence rate of 90% in the general population, and it is more frequently observed among the pediatric population [10]. Leg length discrepancy can be measured clinically by measuring the length from the anterior superior iliac spine to the medial malleolus and calculating the difference between the two lower limbs. Another method of measurement is to calculate the difference in length between the two malleoli in the supine position. A more precise method is to measure the difference in length on a radiograph of the lower extremities under load. It is possible to measure the length by calculating the difference in the height of the iliac crest or the femoral heads on a radiograph of the pelvis or by measuring the lengths of the tibia and femur. The femur is measured from the top of the greater trochanter to the most distal point of the lateral condyle. The tibia is measured from the most proximal point to the most distal point at the ankle joint line.

Length differences are typically less than 10 mm, asymptomatic, and develop as a momentary condition during growth. In some rare cases, children are born with leg discrepancies, while other causes are acquired (fractures, tumors, radiation, infections). LLI is classified as mild (0–2.5 cm), moderate (2.5–6 cm), and severe (>6 cm) [11] and can be categorized etiologically as structural or functional. Structural or anatomical LLI is due to the physical shortening or lengthening of a unilateral lower extremity, while functional LLD refers to the apparent asymmetry of the lower extremity, without the physical shortening or lengthening of the osseous components of the lower limb. A functional leg length discrepancy (LLD) refers to a situation where one leg appears longer than the other due to factors such as pelvic tilt, muscular imbalances, or poor alignment, rather than an actual difference in bone length. Unlike a structural LLD, where there is a measurable difference in the bones’ length, a functional LLD is often temporary and can be corrected with proper intervention. If the pelvis is tilted or rotated, it can affect the apparent leg lengths. This can occur due to muscle imbalances, joint issues, or posture problems. Tightness or weakness in the hip, thigh, or calf muscles can lead to altered alignment and functional LLD. Tightness in soft tissues, such as ligaments and fascia, can contribute to uneven alignment of the pelvis and legs. Unlike structural LLD, functional LLD cannot be corrected with a lift, but it requires physical therapy to assess posture, muscle imbalances, and alignment so as to develop a personalized exercise program that addresses the underlying causes of functional LLD [7]. The effects of LLD on the spine vary depending on the cause and size of the difference. The correlation between LLI, the alignment of the spine, and pelvic imbalance has been assessed in various ways, even methods based on simulating LLI [12] and studying its consequences for trunk, spinal, and pelvic posture [13,14,15]. These parameters regress with the equalization of LLI [16]. Scoliosis due to LLD is referred to as functional scoliosis, and it totally or partially regresses when the LLD is eradicated.

The pattern of scoliosis associated with LLD is described as compensatory, non-structural, and non-progressive, but it has been suggested that LLD can produce structural changes in the spine over time.

LLD can also occur secondary to scoliosis, particularly in the case of compensatory scoliosis. In these cases, LLD appears as the result of an asymmetrical load on the lower extremities. However, the factors associated with variations in LLD and its relationship with pelvic obliquity are unknown [9]. Moreover, the literature provides discordant results on the degree of LLI that can cause vertebral misalignment. Some authors believe that an LLD of 5 mm or less has real significance for mechanically related dysfunctions around the hips, pelvis, and spine, while other investigators believe that an LLD of less than 1 cm is not significant and has no pathological implication [11,12,13,14,15,16,17].

No study has ever investigated the specific relationship between LLI, sacral inclination, and vertebral rotation in patients of growing age with functional scoliosis (Cobb < 10°). As previously mentioned, the most frequent type of scoliosis is “adolescent idiopathic scoliosis (AIS)”. Its cause is unknown, and the prognostic factors linked to curve progression are still debated. Prognosis can vary widely depending on factors such as the severity of the curvature, the age of onset, the underlying cause, and the patient’s overall health.

The severity of the curvature (degree of spinal curvature measured based on the Cobb angle on radiograph) is a crucial factor in predicting the prognosis. Mild curves (less than 20–25 degrees) are generally considered as less likely to progress significantly, while more severe curves may have a higher likelihood of progression.

Age of onset can influence prognosis. Early onset scoliosis, occurring before puberty, tends to have higher potential for progression due to growth spurts during adolescence. Skeletal maturity is another important prognostic factor; once growth is complete, the progression of scoliosis usually slows down significantly. The greater the stage of skeletal maturity is, the less likely the curvature will progress.

Curve pattern and location: The location and pattern of the curves can impact prognosis, as can gender; in general, girls are more likely to experience scoliosis progression than boys, especially during growth spurts. This is particularly true for idiopathic scoliosis.

Family history: A family history of scoliosis might increase the likelihood of progression, suggesting a genetic predisposition. It is not known which has the greatest influence on prognosis.

Recently, many studies have demonstrated the importance of vertebral rotation. Indeed, the maintenance of the viscous–elastic property of the intervertebral disc depends on this aspect, which is directly linked, together with the Cobb degree, to the distribution of forces on the spine. In children, especially, the spine is in a state of dynamic equilibrium; the entire spine is subject to elastic deformation during movement and has the ability to return to its primitive configuration. It has been demonstrated that when the column starts in an altered condition, it imposes alterations of movement followed by a change in elastic return, which can lead to structural changes over time [18]. These alterations have a great impact on the evolution of the scoliotic curve. The resetting of vertebral rotation has been shown to change the progression of the curve once conservative treatment has ended [19]. Based on these premises, we decided to focus our attention on vertebral rotation in curves of less than 15 degrees in patients of growing age with LLI.

The aim of this study was to research the presence of vertebral rotation in functional scoliosis caused by limb length inequality (LLI). In addition, we aimed to examine the correlation between LLI, sacral inclination, and vertebral rotation to discover whether there is a quantitative measure of LLI in which the risk of vertebral body rotation increases.

## 2. Materials and Methods

### 2.1. Design of the Study

This study was a retrospective analysis of 343 consecutive patients (male and female) who underwent view-standing X-rays of the whole spine in our hospital from September 2022 to November 2022. We only selected X-rays from our hospital database, featured in our Carestream program (Figure 1).

### 2.2. Population

The inclusion criteria were a Cobb angle < 15°, primary scoliosis, the absence of thoracic curve, sacral shelf, femoral heads visible on X-ray, and age less than 16 years old. The exclusion criteria were curves secondary to other pathologies, thoracic curves, or combined curves, curves with a Cobb angle > 15°, sacral shelf, and femoral heads that were not visible. From among 343 patients, we found 89 patients meeting the inclusion criteria. In the X-ray of the spine in two projections, we measured the presence of a vertebral rotation seat of the curve, as well as the entity of the curve using the Cobb degree, observed the Risser degree, and determined whether the scoliosis was primary or secondary to bone causes (Figure 2A,B). We also measured limb length inequality (LLI) and sacral shelf inclination. The femoral horizontal reference line was defined as a horizontal line tangent to the top of the highest part of the femoral head. The height between the right and left femoral horizontal reference lines was defined as the size of the LLI. The inclination of the sacral shelf was measured by drawing a horizontal line at the level of the first two foramina or the sacroiliac joint, while vertebral rotation was evaluated using a Perdriolle’s torsionmeter (Figure 3). To render the measurements more precise, in addition to the X-ray grid reference, we double-checked the measurement using a ruler placed on the computer screen. During the visit of the patient with suspected scoliosis, we evaluated the symmetry of the sacral shelf and measured the length of the lower limbs. It is important to observe the change in the alignment of the spine by applying a lift below the limb, showing a measure inferior to the contralateral limb (Figure 4).

All measurements were performed by a single operator.

### 2.3. Statistical Analysis

The statistical analysis was performed using STATA (Stata, College Station, TX, USA), and a *p* value less than 0.05 was considered statistically significant.

The Shapiro–Francia test was used to check the normality of each variable. Pearson’s correlation coefficient and the logistic regression type (r2) were calculated for the correlation between pelvic inclination, vertebral rotation, and limb length inequality (LLI).

The correlation of pelvic obliquity (SSI) and LLI was calculated for the whole group (N = 89 subjects) and divided by vertebral rotation (Perdriolle 0° group N = 38; Perdriolle 5–15° group N = 51).

## 3. Results

Fifty-seven out of eighty-nine patients had vertebral rotation (51% of cases) of the apical vertebrae. Most cases involved L2 (21 out of 57 cases). The Risser sign was 1.8 ± 1.9 (mean ± SD). A total of 33 patients had a left lumbar curve, 36 had a left dorso-lumbar curve, 3 had a right lumbar curve, and 17 had a right dorso-lumbar curve (Table 1).

In these patients, vertebral rotation ranged from 5 to 15 Peridiolle’s degrees. In total, 25 showed 5 degrees of rotation, 27 showed 10 degrees of rotation, and 5 showed 15 degrees of rotation (Table 2). Seven patients did not show lower limb inequality, and of these, only one showed vertebral rotation.

Fifty-eight patients had between 5 mm and 24 mm of LLI, while the remaining patients had a lower degree of LLI. Of these 58 patients, only 15 did not show vertebral rotation, while among the 24 patients with LLI of less than 5 mm, 14 not show vertebral rotation (Table 3).

The mean LLI value of the whole group was 6.5 ± 4.59 mm. Splitting the patients into two subgroups, those without rotation and those with rotation, the mean LLI value was 5.2 ± 4.87 mm for the first group and 7.4 ± 4.18 mm for the second group.

The mean value of sacral shelf inclination (SSI) for the whole group was 2.8 ± 2.53 mm, with a value of 1.4 ± 2.00 mm for patients without rotation and 3.9 ± 2.39 mm for patients with Perdriolle ranging from 5 to 15°. The correlation between sacral inclination and LLI showed a *p* > 0.001 in both subgroups, with rotation and without rotation (Table 4).

The correlation between the inclination of the sacral shelf and vertebral rotation, variables in a statistical relationship of the logistic regression type, showed an R2 value of 0.22 and a *p* < 0.001; both were statistically significant (Figure 5A).

Furthermore, we found a predictive probability according to which, for each millimeter of inclination of the sacrum, it is possible to predict a rotation of the vertebral body of 0.58 degrees, and we found that with a threshold value of 2 mm of inclination, the probability of developing a rotation exponentially increases.

Moreover, the predictive probability of vertebral rotation is 0.23 in the absence of obliquity of the sacral shelf, while it is 0.99 for 11 mm of sacral inclination (Figure 5B).

Instead, the relationship between LLI and vertebral rotation, also variables in a statistical relationship of the logistic regression type, showed an R2 of 0.08, a statistically significant value but one that is smaller than that of the relationship between the sacral shelf and vertebral rotation, for which the *p* value was 0.0016. This value is statistically significant but lower than that of the previous correlation (Figure 6A).

We found that 5 mm is the value of LLI that increases the risk of vertebral rotation. Moreover, the probability of being in the patient group without spinal rotation was 0.38 for patients without heterometry, whereas it increased to 0.93 in the case of heterometry equal to 26 mm (Figure 6B).

The correlation between sacral inclination and LLI was a linear-regression-type statistical relationship and showed a value of *p* < 0.001 (Figure 7).

We found that every mm of length leg inequality corresponds to 0.3 mm of sacral shelf inclination; therefore, vertebral rotation is very likely to occur when LLI reaches a threshold of 5 mm.

## 4. Discussion

According to the literature, to define a curve as “scoliosis”, the deformity in the coronal plane must be greater than 10° Cobb, and it must present vertebral rotation in the transverse plane [20,21,22,23]. Instead, there is a consensus that “functional scoliosis” is an asymmetry in the coronal plane without evidence of a thoracic hump or lumbar asymmetry based on Adam’s test. Often, this alteration is due to limb length discrepancy; it is not progressive and can be corrected without weight bearing [4,5,6,7,8,9,10,11,12,13,14,15,16,17,18,19,20,21,22,23,24]. The results of the present study demonstrated the presence of vertebral rotation in patients with functional scoliosis caused by LLI, and it was correlated with 5 mm of LLI, which will create changes in vertebral and sacral alignment. Previous studies investigated the relationship between LLI and spinal posture with conflicting findings. To the best of our knowledge, this is the first study focusing on vertebral rotation in pediatric patients with functional scoliosis determined by LLI. Moreover, we demonstrated a correlation between LLI, SSI, and vertebral rotation. These results are important, enabling us to better understand the role of vertebral rotation, a parameter related to the progression of this disease [12,13,14,15,16,17,18,19,20,21,22,23,24,25].

Hoikka et al. [12], in a study of 100 patients with an average leg length inequality of 5 mm and a main age of 47 years, found a correlation between LLI and sacral inclination but no relationship between LLI and the Cobb degree. Specht et al., in a retrospective study of 106 consecutive routine diagnostic X-ray procedures, found that 60% of the patients had LLI > 3 mm, 40% had LLI > 6 mm, 50% of the latter had lumbar scoliosis, and only 30% of the first group had lumbar scoliosis [26]. Gibson et al. [27], in a study of patients with LLI ranging from 15 to 55 mm due to a femoral shaft fracture sustained after skeletal maturity, observed that functional scoliosis resolved nearly completely after correction of the leg length discrepancy. However, in this study, the patients showed a contradictory increase in lateral flexion of the column to the shortest leg, although the spine returned to symmetry after LLD correction. This finding contradicts the study of Papaioannou et al., which only included patients who had LLI since childhood (the patients were young adults, and their LLI ranged from 1.2 to 5.2 cm) [28]. These results suggest that a long period of functional scoliosis may result in permanent biomechanical changes in the lumbar spine. The period for which the spine is subjected to functional scoliosis also seems to affect the risk of degenerative changes. Manganiello et al. conducted two different studies to analyze the impact of LLI on the lumbar column, and they even suggested that low LLI can induce high desalination of the lumbar region with respect to major LLI (>2 cm) [29,30]. They also proposed that LLI could be the primum movens for the onset of structured scoliosis. These findings supplemented those of the aforementioned studies demonstrating that changes in spinal alignment can form over time, suggesting a possible structuring of vertebral rotation over time secondary to the difference in length of the lower limbs. Although the recent literature has shown a relationship between the LLI and lumbar scoliosis, Grivas et al. analyzed patients with LLI ranging from 0.5 cm to 2 cm and found that LLI was significantly correlated with the 4DF (4D Formetric DIERS apparatus) reading of pelvis rotation, pelvic tilt, and surface rotation, while it was not correlated with the scoliosis angle or the scoliometer reading at the lumbar level [31]. Instead, Betsch et al. simulated LLI > 2 cm in 100 volunteers (53 females and 47 males) with a mean age of 34 years, finding a correlation between LLI, pelvic inclination, surface rotation, and lateral inclination (all parameters were investigated with raster stereography). In a previous study, the authors did not observe postural impairment for LLI < 2 cm [13,14].

Furthermore, a relationship between leg length inequality and adolescent idiopathic scoliosis (AIS) was also demonstrated. In a recent study published in the *Asian Spine Journal*, Kobayashi et al. [9] demonstrated a direct relationship between LLI, the Cobb angle, and vertebral rotation in 23 patients with AIS. A correlation was found between LLI and vertebral rotation, but compared to our study, the number of patients was small and included scoliosis patients with a Cobb angle between 10 and 30 degrees and LLI > 2 cm. Sekiya et al. [32] found a correlation between functional LLI, pelvic obliquity, and the Cobb angle of the lumbar region, but they suggested that in this case, LLI was secondary to AIS. This study revealed that patients with AIS have functional LLD but not significant structural LLD. The authors reported that the relationship between the lumbar Cobb angle and functional LLD indicates that the lumbar curve contributes to functional LLD; thus, the difference between functional and structural LLDs represents a compensatory mechanism involving the extension and flexion of the lower limbs. None of these studies specifically focused on the consequences of LLI for both the alignment of the sacral shelf and the lumbar spine, exploring how these affects vertebral rotation. Moreover, patients affected by idiopathic AIS show a rotation of the pelvis and the sacrum in addition to an inclination, and it has been demonstrated that these pathologies can arise because of an LLI. In the radiographs of patients affected by scoliosis, the right ilium often appears to be wider than the left ilium in patients with major thoracic curves, while the left ilium often appears to be wider than the right ilium in patients with major thoracolumbar/lumbar curves. Gum et al. also noted this phenomenon and interpreted it as the result of transverse pelvic rotation. They suggested that the transverse plane pelvic position that accompanies major thoracic curves is the fourth transverse plane compensation. The direction of transverse pelvic rotation is the same as that for the main thoracic curve in most patients with a compensatory thoracolumbar/lumbar curve [15,16,17,18,19,20,21,22,23,24,25,26,27,28,29,30,31,32,33,34]. These studies can, therefore, explain the relationship that we found in our study between leg length inequality, sacral shelf, and vertebral rotation.

Our study was limited by the difficulty of undertaking a differential diagnosis between structured scoliosis and functional scoliosis due to LLI when there is vertebral rotation, because most lumbar curves are not progressive and the prognostic factors and causes of AIS are not yet known.

Our future objectives will be to follow these patients up to skeletal maturity, to observe the evolution of functional scoliosis due to LLI, and to understand the role of vertebral rotation. Another interesting point to evaluate is whether the use of a custom foot orthosis with sole lift would be useful in cases of a discrepancy starting from 5 mm to avoid the onset of a possible rotation that could not be reduced over time.

## 5. Conclusions

Functional scoliosis due to leg length inequality can involve vertebral rotation with a direct correlation between leg length inequality, sacral shelf inclination, and vertebral rotation. A limb length inequality greater than 5 mm can be considered as the threshold value above which the sacral shelf could tilt, causing a rotation of the spine.

## Figures and Tables

**Figure 1 jcm-12-05571-f001:**
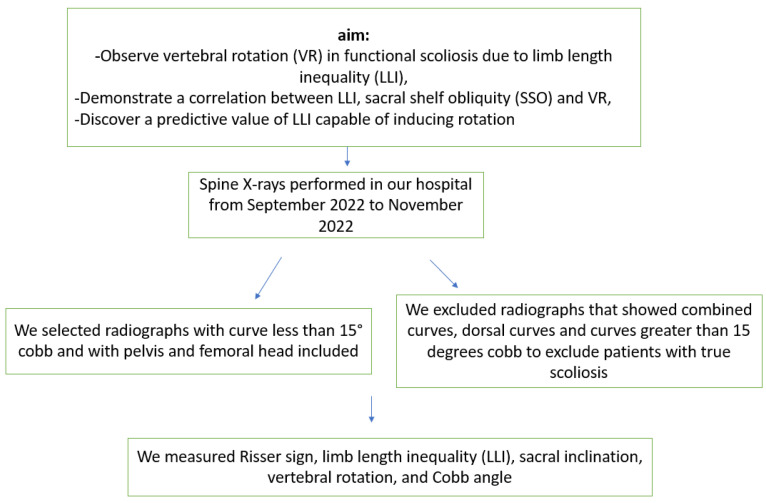
Flow chart of the study design.

**Figure 2 jcm-12-05571-f002:**
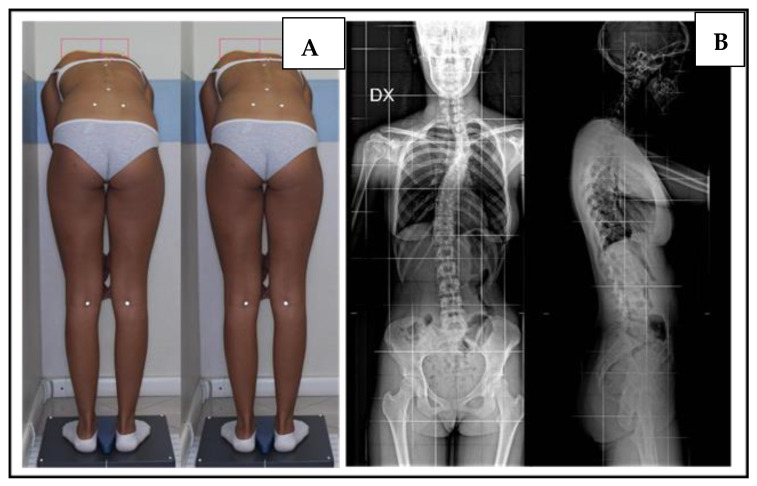
(**A**) Clinical aspects of scoliosis; (**B**) Radiological aspects of scoliosis.

**Figure 3 jcm-12-05571-f003:**
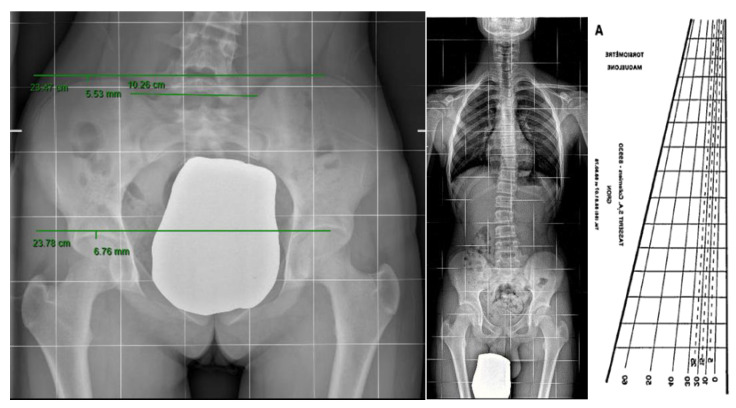
These figures show how the measurements were taken. A: Pedriolle’s Torsionometer.

**Figure 4 jcm-12-05571-f004:**
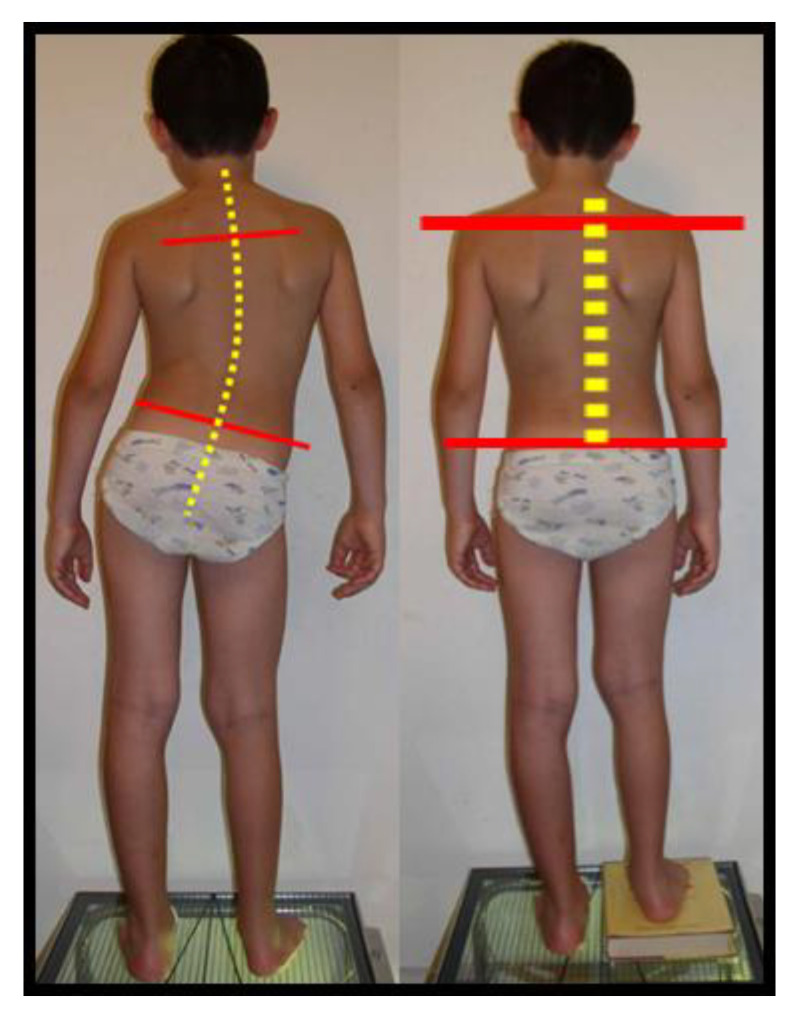
Clinical aspects of functional scoliosis.

**Figure 5 jcm-12-05571-f005:**
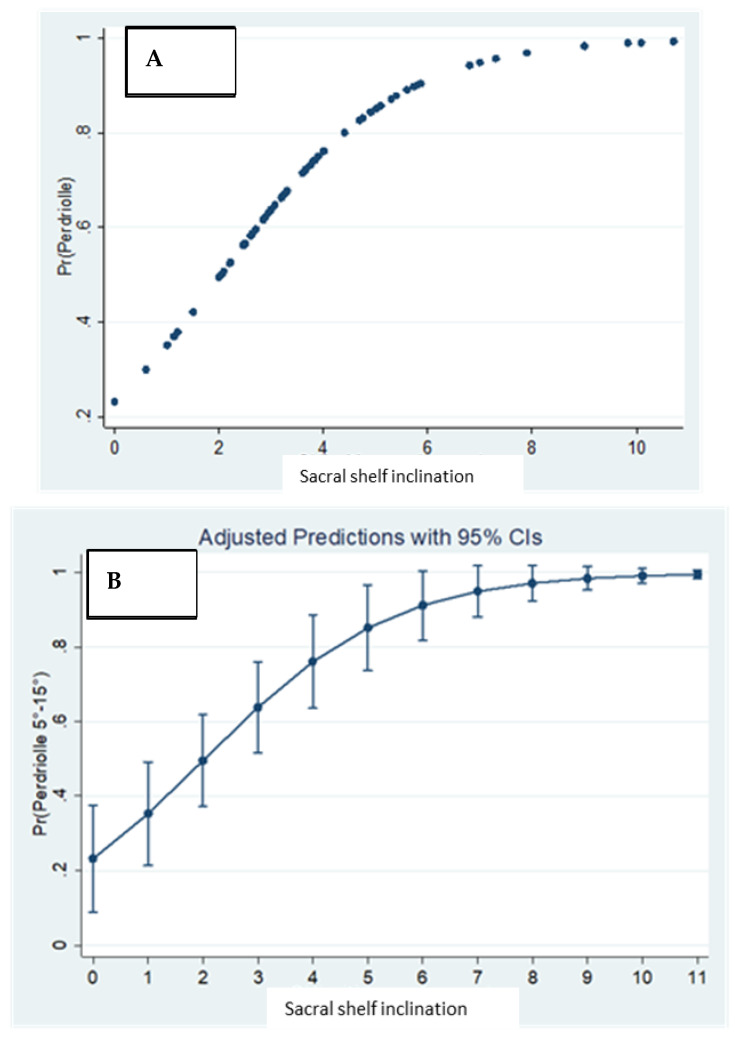
(**A**) Relationship between sacral shelf inclination and vertebral rotation. (**B**) Predictive probability of vertebral rotation correlated with sacral shelf inclination.

**Figure 6 jcm-12-05571-f006:**
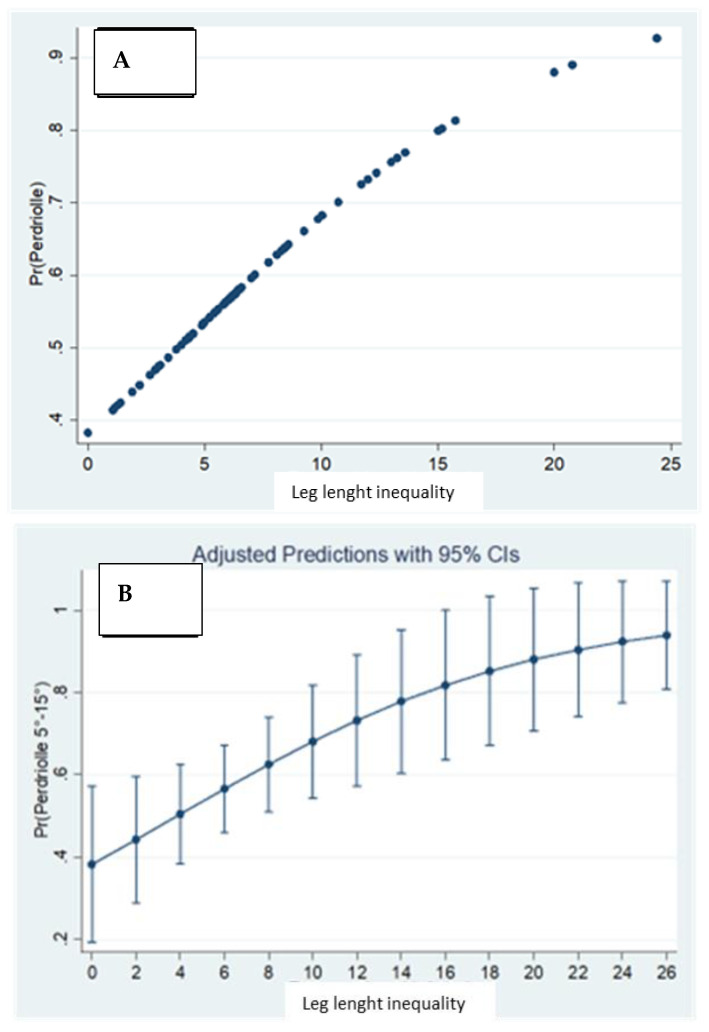
(**A**) Relationship between LLI and vertebral rotation. (**B**) Probability of being in the patient group without or with vertebral rotation based on LLI.

**Figure 7 jcm-12-05571-f007:**
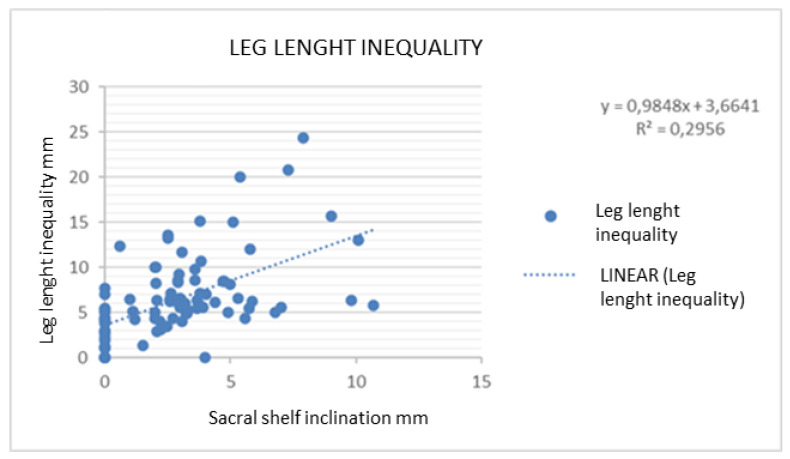
Relationship between LLI and sacral shelf inclination.

**Table 1 jcm-12-05571-t001:** Curve location and n° of patients.

Curve Location	n° Patients
Left lumbar	33
Left dorso-lumbar	36
Right lumbar	3
Left dorso-lumbar	17

**Table 2 jcm-12-05571-t002:** Perdriolle’s value and n° of patients.

Perdriolle	N° Patients
5	25
10	27
15	5

**Table 3 jcm-12-05571-t003:** LLI, n° of patients and rotation.

	LLI 5–24	LLI < 5 mm	LLI 5–24 with Rotation	LLI < 5 mm with Rotation
N° Patients	58	24	43	10

**Table 4 jcm-12-05571-t004:** Mean values of lower limb inequality (LLI) and sacral shelf inclination (SSI) in the whole group and in patients with and without rotation.

	LLI (Lower Limb Inequality) Mean ± SD	SSI (Sacral Shelf Inclination) Mean ± SD	Cohen d	*p* Value
Total sample	6.5 ± 4.59	2.8 ± 2.53	0.97	*p* < 0.001
With Rotation	7.4 ± 4.18	3.9 ± 2.39	1.03	*p* < 0.01
Without rotation	5.2 ± 4.87	1.4 ± 2.00	0.94	*p* < 0.001

## Data Availability

Datasets generated and/or analyzed during the current study are available from the corresponding author upon reasonable request.

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
