# Peer review of "Vertebral Rotation in Functional Scoliosis Caused by Limb-Length Inequality: Correlation between Rotation, Limb Length Inequality, and Obliquity of the Sacral Shelf"

_jcm, 2023, doi:10.3390/jcm12175571_

Round 1

Reviewer 1 Report

Dear Authors,

I would like to congratulate yourself for the developed research. The theme of your study is highly relevant. Please consider the appointments below.

11)    The manuscript needs a review for an English native speaker.

22)      Some references are incomplete. Please, review the references 1, 9, 26, 32, 33, and 37.

33)     The conclusion must answer only the purpose of the study both in the abstract and at the end of the manuscript. Which results of the present study could support that the correction of LLI can avoid the onset of a possible non-reducible rotation?

44)     Please, describe the acronym LLD when it first appears in the text.

55)     Why the absence of thoracic curve was one of the inclusion criteria adopted?

66)     In the results, I missed the p-value in the comparison between the group vertebral rotation (VR) and without vertebral rotation (WVR), especially in table 4. In addition, I suggest standardizing the acronyms VR and WVR for the groups.

77)     Tables 1 and 2 are not necessary because they only repeat the text information.

88)      On page 5, in the lines 184 and 185, it is necessary to add the unit of measure.

99)      In the table 4, the data for “Mean value LLI (no rotation) is the same of “Mean value SSI”. It is necessary to improve the presentation of data in the table. For example, what means the number after ±?

110)   I suggest that the first paragraph of the discussion talks about the main results of the present study instead of the scoliosis´ definition.

111)  In the discussion, the authors said that this is “…the first study focused on vertebral rotation in pediatric patients…”. But, there is no data about the age of the sample studied. Please, include this information. Is there some age group in the inclusion criteria adopted by the present study?

112)  Please, describe the acronym 4DF (page 9, line 297)

Author Response

Reviewer reports:

Reviewer #1: I would like to congratulate you for the developed research. The theme of your study is highly relevant. Please consider the appointments below.

1)    The manuscript needs a review for an English native speaker.

-the text was revised

2)      Some references are incomplete. Please, review the references 1, 9, 26, 32, 33, and 37.

- We have completed the references

3)     The conclusion must answer only the purpose of the study both in the abstract and at the end of the manuscript. Which results of the present study could support that the correction of LLI can avoid the onset of a possible non-reducible rotation?

-This second conclusion is a guess on the basis of the fact that we have shown a statistically significant relationship between mm of heterometry, those of shelf inclination, and the amount of rotation, therefore an LLI can cause a lumbar curve with an associated rotation that could structure over time, this will then have to be demonstrated in subsequent studies. But we have remove the sentence from conclusion and we add it in discussion (line 341 )

4)     Please, describe the acronym LLD when it first appears in the text.

-We corrected this in the text (line 74)

5)     Why the absence of thoracic curve was one of the inclusion criteria adopted?

-since in the literature it has been demonstrated that only the lumbar curves can be secondary to LLI and also to exclude cases of true scoliosis which would have made the result less reliable

6)     In the results, I missed the p-value in the comparison between the group vertebral rotation (VR) and without vertebral rotation (WVR), especially in table 4. In addition, I suggest standardizing the acronyms VR and WVR for the groups.

- The p value was reported only in the text and was showed in the figure.

7)     Tables 1 and 2 are not necessary because they only repeat the text information

-  In our opinion, the table are more simple to read, and I prefer repeat it.

8)      On page 5, in the lines 184 and 185, it is necessary to add the unit of measure.

-We corrected it in the text

9)      In the table 4, the data for “Mean value LLI (no rotation) is the same of “Mean value SSI”. It is necessary to improve the presentation of data in the table. For example, what means the number after ±?

-We corrected it in the text and we have changed the table  4

10)   I suggest that the first paragraph of the discussion talks about the main results of the present study instead of the scoliosis´ definition.

- We don’t remove the first paragraph but we add our results (line 283-291)

11)  In the discussion, the authors said that this is “…the first study focused on vertebral rotation in pediatric patients…”. But there is no data about the age of the sample studied. Please, include this information. Is there some age group in the inclusion criteria adopted by the present study?

-Our sample has a maximum age of 16 years.  We add in the inclusion criteria the age. We add in the results the Risser mean value.

12)  Please, describe the acronym 4DF (page 9, line 297)

added in the text.

Reviewer 2 Report

The manuscript is well-written overall, but some issues must be addressed.

-Choose what type of paper it is

- Please correct the rows in the abstract part

-please state where the pictures are taken from ( personal collection, other materials with permission..)

-please insert a flow chart regarding the study design.

-what were the exclusion criteria?

-please expand the methods part

-please state the alpha value

-also include a posthoc  power analysis

Author Response

Reviewer #2: The manuscript is well-written overall, but some issues must be addressed.

-1Choose what type of paper it is

-it is an article, I added in the manuscript

-2 Please correct the rows in the abstract part

-We corrected the rows

-3please state where the pictures are taken from ( personal collection, other materials with permission..)

-The picture are taken from personal collection

- 4 please insert a flow chart regarding the study design.

- We added in the text

-5 what were the exclusion criteria?

-Exclusion criteria were curves secondary to other pathologies, thoracic curves, or combined curves. We added in the text

- 6 please expand the methods part

-We expanded it

- 7 please state the alpha value

-If you intend the P>value it is in  the text p<0.05 (Statistical analysis)

- 8 also include a posthoc  power analysis

-Post hoc power analysis in linear regression should not be relied upon as a valid approach to assess the quality of a study's design or to interpret its results.